# Dietary Use of Methionine Sources and *Bacillus amyloliquefaciens* CECT 5940 Influences Growth Performance, Hepatopancreatic Histology, Digestion, Immunity, and Digestive Microbiota of *Litopenaeus vannamei* Fed Reduced Fishmeal Diets

**DOI:** 10.3390/ani13010043

**Published:** 2022-12-22

**Authors:** Ramón Casillas-Hernández, Jose Reyes Gonzalez-Galaviz, Libia Zulema Rodriguez-Anaya, Juan Carlos Gil-Núñez, María del Carmen Rodríguez-Jaramillo

**Affiliations:** 1Departamento de Ciencias Agronómicas y Veterinarias, Instituto Tecnológico de Sonora, 5 de Febrero 818 Sur, Colonia Centro, Cd. Obregón 85000, Sonora, Mexico; 2CONACYT-Instituto Tecnológico de Sonora, 5 de Febrero 818 Sur, Colonia Centro, Cd. Obregón 85000, Sonora, Mexico; 3Centro de Investigaciones Biológicas del Noroeste, Calle Instituto Politécnico Nacional No. 195, La Paz 23096, Baja California Sur, Mexico

**Keywords:** *Litopenaeus vannamei*, shrimp nutrition, fishmeal replacement, methionine, probiotics, performance, health, microbiota

## Abstract

**Simple Summary:**

The accelerated expansion of shrimp farming requires protein sources with high nutritional value to formulate feeds that satisfy shrimp nutritional requirements. Fishmeal (FM) is the main protein source for aquafeed formulations. However, its limited supply and high cost encourage research on alternative protein sources to formulate more profitable feeds that contribute to aquaculture sustainability. Soybean meal (SBM) and poultry by-product meal (PBM) have been used as protein sources for replacing fishmeal, but their essential amino acids imbalance contributes to low shrimp growth performance and affect shrimp health. Therefore, the study purpose was to evaluate the effect of FM replacement by SBM and PBM in diets supplemented with DL-Met, MET-MET (AQUAVI^®^), *Bacillus amyloliquefaciens* CECT 5940 (ECOBIOL^®^) and their combinations on growth performance and health of juvenile *Litopenaeus vannamei*. The results showed that FM could be partially replaced with SBM and PBM in shrimp feeds supplemented with 0.19% MET-MET or 0.06% MET-MET plus 0.10% *B. amyloliquefaciens* CECT 5940 without adversely affecting the growth performance and welfare of *Litopenaeus vannamei*. These results may be interesting for developing low fishmeal feeds and contributing to aquaculture sustainability.

**Abstract:**

An 8-week feeding trial investigated the effect of Fishmeal (FM) replacement by soybean meal (SBM) and poultry by-product meal (PBM) in diets supplemented with DL-Met, MET-MET (AQUAVI^®^), *Bacillus amyloliquefaciens* CECT 5940 (ECOBIOL^®^) and their combinations on growth performance and health of juvenile *Litopenaeus vannamei*. A total of six experimental diets were formulated according to *L. vannamei* nutritional requirements. A total of 480 shrimp (0.30 ± 0.04 g) were randomly distributed into 24 tanks (4 repetitions/each diet, 20 shrimp/tank). Shrimp were fed with control diet (CD; 200 g/Kg fishmeal) and five diets with 50% FM replacement supplemented with different methionine sources, probiotic (*B. amyloliquefaciens* CECT 5940) and their combinations: D1 (0.13% DL-MET), D2 (0.06% MET-MET), D3 (0.19% MET-MET), D4 (0.13% DL-MET plus 0.10% *B. amyloliquefaciens* CECT 5940 and D5 (0.06% MET-MET plus 0.10% *B. amyloliquefaciens* CECT 5940). Shrimp fed D3 and D5 had significantly higher final, weekly weight gain, and final biomass compared to shrimp fed CD (*p* < 0.05). Shrimp fed D2 to D5 increased the hepatopancreas epithelial cell height (*p* < 0.05). Digestive enzymatic activities were significantly increased in shrimp hepatopancreas’ fed D3 (*p* < 0.05). Meanwhile, shrimp fed D1 had significant downregulation of immune-related genes (*p* < 0.05). Moreover, shrimp fed D3 and D5 increased the abundance of beneficial prokaryotic microorganisms such as *Pseudoalteromonas* and *Demequina* related to carbohydrate metabolism and immune stimulation. Also, shrimp fed D3 and D5 increased the abundance of beneficial eukaryotic microorganism as *Aurantiochytrium* and *Aplanochytrium* were related to eicosapentaenoic acid (EPA) and docosahexaenoic acid (DHA) production which plays a role in growth promoting or boosting the immunity of aquatic organisms. Therefore, fishmeal could be partially substituted up to 50% by SBM and PBM in diets supplemented with 0.19% MET-MET (AQUAVI^®^) or 0.06% MET-MET (AQUAVI^®^) plus 0.10% *B. amyloliquefaciens* CECT 5940 (ECOBIOL^®^) and improve the productive performance, health, and immunity of white shrimp. Further research is necessary to investigate synergistic effects of amino acids and probiotics in farmed shrimp diets, as well as to evaluate how SBM and PBM influence the fatty acid composition of reduced fishmeal diets and shrimp muscle quality. Nevertheless, this information could be interesting to develop low fishmeal feeds for aquaculture without affecting the growth and welfare of aquatic organisms.

## 1. Introduction

Shrimp farming yielded almost 11.2 million tons in 2022 and Pacific white shrimp (*Litopenaeus vannamei*) was the most representative species with 52% of total production [1]. The sustainability and profitability of aquaculture production requires the supply of raw materials with high nutritional value for feed formulation that meet the nutritional needs of farmed shrimp [2,3].

Fishmeal is the main ingredient in aquafeed due to its high nutritional value [4]. However, the limited supply and high cost of fishmeal require research into alternative sources for feeds that are more profitable, that contribute to aquaculture sustainability [5,6].

Previous studies, mainly focused on shrimp productive performance, reported that fishmeal can be partially substituted by various animal and vegetable sources [6]. Among vegetable protein sources, soybean meal has a high protein content. However, the presence of anti-nutritional factors, poor digestibility, and an essential amino acids (EAA) imbalance affecting digestive microbiota, that causes an inflammatory response in digestive organs, low productivity performance, and affects the aquatic organism’s immune response [5,7,8]. Poultry by-product meal is a high protein animal source deficient in methionine and lysine, therefore, its use in reduced fishmeal feeds could affect growth and welfare of aquatic organisms [9].

Methionine is an EAA scarce in low fishmeal aquafeeds and is necessary for normal growth [10], protein synthesis [11], and immune function [12] of aquatic organisms. Therefore, methionine supplementation in low fishmeal feeds is required to balance amino acids and reduce negative impacts on growth and metabolism of farmed aquatic organisms [13]. A variety of methionine resources are commercially available, such as a racemic mixture of D-Met and L-Met isomers called DL-methionine (DL-Met) and a mixture of four different methionine stereoisomers (LD-Met-Met, DL-Met-Met, LL-Met-Met and DD-Met-Met) commercialized as AQUAVI^®^ (Met-Met) [10]. However, AQUAVI^®^ has better physicochemical properties such as very low water solubility and higher absorption than DL-Met [10,14].

On the other hand, it has been reported that probiotics (*Bacillus subtilis*) supplementation has positive effects on growth performance and health in white shrimp (*L. vannamei*) and bullfrog (*Lithobates catesbeianus*) fed low fishmeal feed [15,16]. Probiotics are live microorganisms used in aquafeeds due to their capacity for improving feed utilization, enzymatic digestion, pathogen prevention, immune response, and growth [15]. Therefore, the probiotics’ inclusion in reduced fishmeal feeds could improve health and welfare of aquatic organisms [16]. *Bacillus* bacteria are widely used in aquaculture as probiotics because they have the capacity to produce antimicrobial compounds and exoenzymes that improve nutrient digestion, pathogen inhibition, immune response modulation, gut integrity maintenance, and consequently the growth performance [17]. *B. amyloliquefaciens* has antibacterial activity and produces digestive enzymes that support digestion [18].

Due to the above, alternative sources to fishmeal, combined with many additives have been used in aquafeeds to guarantee an essential nutrient supply, improve productive performance, preserve the diet’s physicochemical composition, and maintain the aquatic environment’s quality [19]. Hence, low fishmeal diets supplemented with different methionine sources and *B. amyloliquefaciens* have improved the growth performance and feeding efficiency of farmed aquatic organisms [3,7,8,20,21]. Also, *B. amyloliquefaciens* CECT 5940 and its effects have been reported on growth performance and health of broiler chickens [22] and Nile tilapia (*Oreochromis niloticus*) [21]. However, few studies have validated the effects of low fishmeal diets and additives on the digestive health and immune response of shrimp. Therefore, the main objective of this study was to evaluate the effect of FM replacement with SBM and PBM in diets supplemented with DL-Met, MET-MET (AQUAVI^®^), *B. amyloliquefaciens* CECT 5940 (ECOBIOL^®^) and their combinations on growth performance, hepatopancreatic histology, digestive enzymatic activity, transcriptional response of immune-related genes, and the microbial composition of the digestive system of juvenile Pacific white shrimp (*L. vannamei*).

## 2. Materials and Methods

### 2.1. Experimental Diet Preparation

A total of six experimental diets were formulated according to the *L. vannamei* nutritional requirements [23] and manufacturer recommendations (Evonik’s aqua R & D group). The control diet (CD) was formulated to contain a mixture of FM protein (39%), SBM protein (53%), and PBM protein (8%) like a typical commercial shrimp diet. SBM and PBM were the major alternative protein sources to replace FM, while wheat flour was used as a binder and filler ingredient. Fish oil was used as the major lipid source to satisfy the n-3 essential fatty acid shrimp requirements. CD had 200 g/Kg fishmeal without any supplementation. The other five diets (D1–D5) where SBM and PBM replaced FM at 50% were supplemented with different methionine sources, *B. amyloliquefaciens* and their combinations. D1: 0.13% DL-MET, D2: 0.06% MET-MET (AQUAVI^®^), D3: 0.19% MET-MET (AQUAVI^®^), D4: 0.13% DL-MET plus 0.10% *B. amyloliquefaciens* CECT 5940 (ECOBIOL^®^) and D5: 0.06% MET-MET (AQUAVI^®^) plus 0.10% *B. amyloliquefaciens* CECT 5940 (ECOBIOL^®^). In addition, the water stability ratio was evaluated as a diet’s physical characteristic by a previously described method [24]. The formulation, proximate composition, and water stability values of the experimental diets are shown in Table 1.

### 2.2. Shrimp, Feeding Trial and Sample Collection

Juvenile *L. vannamei* were obtained from a shrimp farm from Sonora, Mexico. The shrimp were pathogen-free according to procedures described in the Manual of Diagnostic Tests for Aquatic Animals of the World Organization for Animal Health [25]. Before the feeding trial, all shrimp were acclimated in the aquaculture laboratory in 1500-L tanks with seawater under controlled conditions (temperature 30 ± 0.5 °C, dissolved oxygen ≥4 mg/L, salinity 37 g/L, pH ≥ 7 and photoperiods of 12 light hours) and fed with a commercial feed for 7 days.

At the start, 480 healthy shrimp with (0.30 ± 0.02 g) were fasted for 24 h and randomly distributed to 24 circular tanks (volume is 150 L) at a density of 20 shrimp per tank (equivalent to a density of 133 shrimp/m^3^). There were four replicate tanks that were randomly assigned for each dietary treatment. Shrimp were fed to satiation with an initial ration of 12% of their biomass divided into three rations during the day (8:00, 13:00 and 16:00 h) for 56 days, adjusted daily depending on residual feed presence or absence. Temperature (27.84 to 28.36 °C), salinity (36.98 to 37.07 g/L), dissolved oxygen (4.00 to 5.17 mg/L), and pH (7.41 to 7.84) were recorded. Every day, 30% of the water was changed. The uneaten feed, feces, molts, and dead shrimp were removed daily. After a feeding trial and a fasting period of 24 h, three shrimp were randomly sampled from each replicate to obtain 400 μL of hemolymph for gene expression analysis [26]. Shrimp previously bled were aseptically dissected to obtain their whole intestines and hepatopancreas, then were stored at −80 °C until digestive enzymatic activity and microbiome analysis. Additionally, twelve (three shrimp/replicate) whole shrimp were randomly sampled from each treatment and fixed in AFA Davidson solution for histological analysis.

### 2.3. Growth Performance

All shrimp were weighed and counted to calculate the growth performance according to the equations reported by previous studies [27,28,29].

Final weight = (Σ Final individual weight)/Final number of shrimps.Weekly weight gain = (Final weight − Initial weight)/Number of weeks.Specific growth rate = 100 × (ln final weight − ln initial weight)/days of experiment.Survival rate = 100 × (Final number of shrimps/Initial number of shrimps).Final biomass = Final weight × Final number of shrimps.Feed intake = Feed Input (dry weight) − Feed collected (dry weight).Feed conversion rate = Feed intake/Final biomass.

### 2.4. Hepatopancreas Histology

The shrimp hepatopancreas samples fixed in Davidson’s solution were processed according to the method described by Bell and Lightner [30]. Histological sections with thickness of 4 μm, were cut using a rotary microtome (LEICA RM2115RT). The tissue staining, the slide observation, and the images’ digitization were realized according to the method described by Casillas-Hernández [31]. The images of tissues were used to measure hepatopancreas’ cell height using the digital image system Image-Pro Premier software v9.0 (Media Cyvernetics Inc., Rockville, MD, USA).

### 2.5. Digestive Enzymes’ Activity

Intestines and hepatopancreas were separately homogenized to 10% ratio with 0.9% saline solution and centrifuged at 3500 rpm for 10 min at 4 °C, and supernatant was immediately analyzed for digestive enzyme assays with a microplate reader (Bio-Rad, Hercules, CA, USA). Protease and lipase activities were assayed by commercial kit (Sigma-Aldrich^®^, Louis, MO, USA). Amylase activity was measured using soluble starch as substrate [32]. The total soluble protein concentration was determined by the principle of protein dye binding using bovine serum albumin as a standard [33]. Assays were all run in three replicate samples. Digestive enzyme activities are expressed as U/mg of protein.

### 2.6. Transcriptional Response of Immune-Related Genes

The hemolymph was centrifuged at 3500 rpm for 10 min at 4 °C to separate the hemocytes from the plasma. The total RNA from hemocytes was extracted with TRIzol Reagent (Invitrogen, Carlsbad, CA, USA) and treated with RNA-free DNase (Promega^®^, Madison, WI, USA). cDNA was synthesized with Total RNA (500 ng) using the ImProm-II™ Reverse Transcription System (Promega^®^) and oligo d(T)20 (T4OLIGO). The cDNA synthesis was realized by reverse transcription at 42 °C for 60 min; then reverse transcriptase was inactivated at 70 °C for 15 min to stop the reaction. The cDNA was diluted with 80 μL of ultrapure water, and 5 μL was used as template for the real time quantitative PCR (qPCR) reaction. Transcriptional response was analyzed from five immune-related genes and *β-actin* as reference gene (Table 2) [34]. The qPCR was conducted on a StepOne Real Time PCR System (Thermo Fisher Scientific) using SensiFAST™ SYBR^®^ Hi-ROX Kit (Bioline™, London, UK). The qPCR conditions were initial denaturation at 95 °C for 10 min, followed by 40 denaturation cycles at 95°C for 15 s, and annealing/extension at 60 °C for 1 min. An analysis of the dissociation curve (60–95 °C) at a temperature transition rate of 0.5 °C/s was performed for each pair of primers. The relative quantification method was used for gene expression analysis according to Rodriguez-Anaya [35] and Casillas-Hernández [36].

### 2.7. DNA Extraction and Sequencing Analysis

The genomic DNA from hepatopancreas and intestines of 12 shrimps per treatment was extracted using Quick-DNA™ Fecal/Soil Microbe (Zymo Research, Irvine, CA, USA), following the manufacturer’s instructions. The DNA concentration was quantified using a NanoDrop 2000 spectrophotometer (Thermo-Fisher Scientific, Waltham, MA, USA), and DNA quality was evaluated by agarose gel electrophoresis (1%, *w*/*v*). Genomic DNA samples were then outsourced to the Microbial Genomics Laboratory (CIAD, Mexico) for DNA library preparation and sequencing using standard Illumina protocols for amplification [37]. Briefly, the V4 variable region of the 16S rRNA and 18S rRNA genes were amplified by PCR with the following primers using Illumina adapters: 16S-V4_515F (5′-GTG CCA GCM GCC GCG GTA A-3′), 16S-V4_806R (5′-TAA TCT WTG GGV HCA TCA GG-3′), 18S-V9_Euk_1391F (5′-GTA CAC ACC GCC CGT C-3′), and 18S-V9_EukBr (5′-TGA TCC TTC TGC AGG TTC ACC TAC-3′). Finally, the amplicons were quantified in Qubit, mixed in an equimolar pool, and sequenced on Illumina Miniseq platform under standard conditions (300 cycles, 2 × 150).

### 2.8. Bioinformatic Analysis

The FASTQ files from paired-end reads were analyzed using the DADA2 package v1.24.0 [38]. The sequence analysis workflow included filtering, dereplication, sample inference, chimera identification, and merging paired end (PE) reads to group them into ASVs (amplicon sequence variants). DADA2 includes the “naïve Bayesian” method using the SILVA databases for both the 16S-V4 region (silva_nr99_v138.1_train_set.fa) and the 18S-V9 region (silva_132.18s.99_rep_set.dada2.fa). Taxonomic information was analyzed with Phyloseq V1.40.0 and the microbiome package v1.18.0 to obtain alpha and beta diversity and ordination values [37]. The beta diversity was conducted based on unweighted UniFrac distance and visualized using a PCoA built with ggplot in R. Finally, multivariate community-level differences between groups were quantified by permutational multivariant analysis of variance (PERMANOVA) [39]. The final ASVs table from 16S sequences was also used as an input for functional metagenomic prediction using PICRUSt [40]. The KEGG pathway content obtained by PICRUSt was normalized and then used to obtain the metagenomic functional predictions at different hierarchical KEGG levels (1, 2 and 3) [41].

Illumina sequencing using primers for the V4 hypervariable region in the 16S rRNA gene yielded 976,065 PE reads of 150 bp corresponding to the intestine and hepatopancreas of *L. vannamei* with an average of 81,338 reads per sample. After the quality filtering process and elimination of chimeras, an average of 58,235 sequences per sample was maintained, equivalent to 71.6%, and were assigned to 709 ASVs. On the other hand, from the V9-18S rDNA sequencing, 871,699 PE reads of 150 bp were obtained with an average of 72,642 per sample. After sequence analysis workflow, reads were reduced by approximately 12.7%. However, when performing the taxonomic identification and ASV grouping, most of the sequences corresponded to host´s (*L. vannamei*) DNA, eliminating the samples from the hepatopancreas. Therefore, the data presented on eukaryotic microbiota characterization corresponds to the microorganisms present in the intestine. The dataset used were 12,970 sequences and were assigned to 43 ASVs.

### 2.9. Statistical Analysis

The growth performance results, hepatopancreas’ cell height, digestive enzymatic activity, and transcriptional response of immune-related genes were evaluated by one-way analysis of variance (ANOVA). If any significance was observed, Tukey’s test was performed for means comparison. Statistical analysis was performed with Statgraphics Centurion XVI. Significance was set at 95% probability levels.

## 3. Results

### 3.1. Growth Performance

Growth performance values are presented in Table 3. Compared with CD, the highest values of growth performance (final weight, weight gain, and final biomass) were observed with shrimp fed D3 and D5, which were significantly higher than shrimp fed D1 (*p* < 0.05), but without statistical difference with shrimp fed D2 and D4 and (*p* > 0.05). The lowest value of feed conversion rate was observed with shrimps fed D3, but no significant difference was found in FCR among all diet treatments (*p* > 0.05). While feed intake significantly (*p* < 0.05) decreased in shrimps fed D1 and significantly increased (*p* < 0.05) in shrimps fed D5. No significant difference was found in survival rate among all diet treatments (*p* > 0.05).

### 3.2. Hepatopancreatic Histology

The shrimp hepatopancreas had a well-organized structure (Figure 1A). Except for shrimp fed D1, all shrimp fed reduced fishmeal diets had higher (*p* < 0.05) height of the hepatopancreas epithelial cells than shrimp fed CD (Figure 1B).

### 3.3. Digestive Enzyme Activity

The effect of different dietary methionine sources and *Bacillus amyloliquefaciens* on digestive enzyme activities of intestine and hepatopancreas from *L. vannamei* fed reduced fishmeal diets are shown in Table 4. Amylase, protease, and lipase activities of hepatopancreas from shrimp fed D3 were significantly higher (*p* < 0.05) than shrimp fed CD, whereas the digestive enzyme activities of intestine were not influenced (*p* > 0.05). In hepatopancreas from shrimp fed D1 the lowest value of amylase and lipase was observed, while in intestine the digestive enzyme activities had the lowest values but no significant differences (*p* > 0.05) were observed compared with shrimp fed CD.

### 3.4. Transcriptional Response of Immune-Related Genes

The transcriptional response of immune-related genes of white shrimp fed reduced fishmeal diets with additives was determined and compared with the shrimp fed CD (Figure 2). Except for shrimp fed D1, all immune-related genes analyzed in this study had higher expression than shrimp fed CD. Hc and pPO were significantly (*p* < 0.05) upregulated in shrimp fed D3 and D5. LGBP was significantly (*p* < 0.05) upregulated in shrimp fed D2 and D3. MnSOD was significantly (*p* < 0.05) upregulated in shrimp fed D2, D3 and D5. HSP60 was significantly (*p* < 0.05) upregulated in shrimp fed D5. The lowest values of transcriptional response of immune-related genes were observed in shrimp fed D1 compared to all dietary treatments.

### 3.5. Digestive Microbiota Analysis and Functional Prediction

The rarefaction curve analysis showed that the observed species per sample was sufficient for both 16S-V4 (Figure 3A) and 18S-V9 sequences (Figure 4A). Regarding alpha diversity, 16S results showed that the Chao1, Shannon, and Simpsons indexes of the hepatopancreas were higher than intestines (Figure 3B). On the other hand, 18S results showed that all alpha diversity indices decreased in shrimp intestines fed low fishmeal diets compared with control (Figure 4B). However, there were no significant differences between dietary treatments in both 16S and 18S results.

The experimental diet’s effects on microbiota structure were determined using UniFrac distance and visualized using PCoA plots. The results revealed clear separation among experimental diets, but without significant differences in prokaryotic communities from intestine or hepatopancreas (Figure 3C) as well as eukaryotic microbiota from intestine (Figure 4C). However, beta diversity analysis showed a clear separation between prokaryotic communities from intestine and hepatopancreas (Figure 3C).

Based on prokaryotic microbiota results, a total of 19 different bacterial phyla were identified. Proteobacteria was the dominant phylum in shrimp intestine, whereas Proteobacteria and Actinobacteria were the dominant phyla in shrimp hepatopancreas. In shrimp intestines fed D3 and D5, an increase in Actinobacteria relative abundance was observed while Bacteroidota relative abundance was slightly decreased. In shrimp hepatopancreas fed D1, D2 and D4, a slight increase in Bacteroidota relative abundance was observed, while in shrimp hepatopancreas fed D3 and D5 Actinobacteria relative abundance was increased (Figure 5A). At the genus level, *Pseudoalteromonas* was the most prevalent in shrimp intestine while *Demequina* was the most prevalent in shrimp hepatopancreas. *Demequina* had a low increase in shrimp intestine fed D3 and D5, *whereas Lysinimicrobium* and *Ruegeria* were increased in shrimp hepatopancreas fed D3 and D5 (Figure 5B). The functional categories results (KEGG level 2) showed that bacterial sequences were associated with cellular processes and metabolism pathways (Figure 5C and Appendix A). Amino acid metabolism, carbohydrate metabolism, cofactors and vitamin metabolism were most abundant in metabolism pathways, whereas cell motility was most abundant in cellular processes.

After removal of sequencing reads from the host´s DNA, three phyla on eukaryotic microbiota were observed. The SAR (Stramenopiles, Alveolate y Rhizaria) phylum was the most prevalent in all shrimp intestines, however an increase in the Opisthokonta phylum was observed in shrimp intestines fed D5 (Figure 6A). At the genus level, *Aplanochytrium* was the most abundant in shrimp intestine fed D2 to D5, but the abundance was notably higher in D5. Uncultured *Alveolate* was enriched in shrimp intestine fed CD. *Aurantiochytrium* was most prevalent in D2 to D4 with higher abundance in D3. Finally, *Ebria* was most abundant in shrimp intestines fed D1 (Figure 6B).

## 4. Discussion

### 4.1. Growth Performance

Previous studies have evaluated the effects of reduced fishmeal diets supplemented with different dietary methionine sources on growth performance of *L. vannamei*, such as shrimp feed with 5% to 10% fishmeal and supplemented with a level between 0.15% and 1.7% of MET-MET (AQUAVI^®^) or 3% of DL-MET [2,3,8,20,42,43]. Nevertheless, a study with similar culture conditions that used juvenile shrimp (*L. vannamei*) with initial weight 0.98 ± 0.02 g, suggested 0.20% MET-MET (AQUAVI^®^) for better growth performance when shrimp were fed reduced fishmeal feed [20]. DL-MET has been proven on juvenile shrimp with initial weight 3.0 g suggesting levels between 0.06–0.30% for good productive response when shrimp were fed reduced fishmeal feed [2]. On the other hand, the effects of *B. amyloliquefaciens* (10^4^ and 10^3^ UFC/mL) dissolved in water of a biofloc system for farmed *L. vannamei* were reported [44,45], but there are no reports about the effects of this probiotic supplemented in reduced fishmeal feed on *L. vannamei* growth performance. Therefore, this study evaluated the effect of FM replacement by SBM and PBM in shrimp diets supplemented with 0.13% of DL-MET (D1), 0.06% of MET-MET (D2), 0.19% of MET-MET (D3), and according to the manufacturer recommendations we used combinations of 0.13% of DL-MET plus 0.1% *B. amyloliquefaciens* CECT 5940 (equivalent to 10^9^ UFC/g) (D4) and 0.06% of MET-MET plus 0.1% of *B. amyloliquefaciens* CECT 5940 (D5). All reduced fishmeal diets supplemented with methionine sources and probiotic showed good water stability values. Therefore, the fishmeal reduced diets evaluated in this study are in accordance with previous reports and all dietary treatments showed a good response in shrimp growth performance. Nevertheless, shrimp fed D1 had lower performance parameters, while shrimp fed D3 and D5 had higher performance parameters. The significant low growth performance in shrimp fed D1 can be related to poor feed intake due to methionine deficiency causing palatability reduction in reduced fishmeal aquafeeds [20]. The high growth performance in shrimp D3 compared with shrimp fed D1 and D2 could be due to the methionine deficiency and source. The above, may be because it has been reported that MET-MET dipeptide has low water solubility and high bioavailability, which can be efficiently utilized by shrimp promoting better growth performance values [2,10,14]. Shrimp fed D5 also had high growth performance even with the same methionine level and source in D2, but with addition of 0.10% *B. amyloliquefaciens* CECT 5940. The high growth performance could be due to the probiotic’s beneficial properties, which include antimicrobial activity and production of α-amylase, cellulase, and protease that increase the nutrients digestibility and absorption [46]. Also, a previous study reported that *B. amyloliquefaciens* is a methionine producer [47] and this could contribute to maintain the balance of this EAA in D5. Moreover, it has been reported that a mixture of feed additives could enhance the effectiveness of growth performance of aquatic organisms [48,49,50] and broilers [51]. Therefore, the 50% FM replacement by SBM and PBM in diets supplemented with 0.19% MET-MET and 0.06% MET-MET plus 0.10% *B. amyloliquefaciens* CECT 5940 could improve the nutrient utilization and consequently the shrimp growth performance. It is important to note that SBM and PBM have high levels of fatty acids, but n-3 long-chain polyunsaturated fatty acids are deficient in these protein sources [52]. In this context, further research is necessary to determine how the use of SBM and PBM influence the fatty acid composition of reduced fishmeal diets and shrimp muscle quality.

### 4.2. Hepatopancreatic Histology

The shrimp hepatopancreas is a digestive organ and plays an important role in digestive enzyme secretion, nutrient transport, storage, and absorption, therefore, its function is key to shrimp growth performance and health [30,53]. However, it has been reported that reduced fishmeal diets can change the structural morphology of digestive organs and impair physiological conditions in aquatic organisms resulting in growth retardation [54]. The hepatopancreatic histology was analyzed as an indicator for shrimp growth performance, health, and nutritional status. Damage in hepatopancreas structure was not observed, the B cells were the most prevalent hepatopancreatic cells and their epithelial height increased significantly in all dietary treatments with the exception of shrimp fed D1, which had a similar response to that of control diet. The results suggest that the hepatopancreas is sensitive to the inclusion of different dietary methionine sources and *B. amyloliquefaciens* in reduced fishmeal diets, increasing B cell, influencing digestive enzyme secretion, nutrient absorption and assimilation, and feed utilization as reported in other studies when used fishmeal alternative sources and additives [53,55,56,57,58]. Moreover, it has been reported that the methionine supplementation could decrease the hepatopancreas alterations due to its deficiency in low fishmeal diets [8]. Therefore, according to the histological analysis, there is no evidence of toxicity caused by reduced fishmeal diets supplemented with different dietary methionine sources and *B. amyloliquefaciens* in *L. vannamei* hepatopancreas.

### 4.3. Digestive Enzyme Analysis

The enzyme digestive activity is a physiological process that improves nutrient digestion and absorption, and therefore, is a key factor for promoting shrimp growth performance [59]. Nevertheless, previous studies have reported that the digestive enzyme activity decreased significantly with reduced fishmeal diets without additive supplementation [60,61]. The digestive enzyme activities were used as an indicator of shrimp digestive function. In the present study, shrimp fed D3 improved (*p* < 0.05) in hepatopancreatic digestive enzyme activity compared between all groups, and no significant differences were found in the intestinal digestive enzyme activity. Overall, low digestive enzyme activities were noticed in both organs from shrimp fed D1. In agreement with these results, methionine supplementation increased digestive enzymes activities in red sea bream (*Pagrus major*) [62], grass carp (*Ctenopharyngodon idella*) [63], white shrimp (*L. vannamei*) [2,3,20], and rohu fish (*Labeo rohita*) [59]. The effects of feed supplemented with *B. amyloliquefaciens* on shrimp digestive enzyme activity have not been reported in *L. vannamei* but it is known that probiotics increase the digestive enzymes’ activity and improve the feed utilization and digestion [64]. In addition, several works evaluated different aquaculture feeds supplemented with additives combined with probiotics and showed an increase in the digestive process. [48,50]. Nevertheless, inconsistencies in the digestive enzyme activities were evidenced when fed broiler chickens with organic acids, probiotics, and combinations, possibly due to the induction level of feed additives and combinations being at suboptimal levels [51]. This study also reported inconsistencies in hepatopancreatic digestive enzyme activity in shrimp fed D3, D4, and D5. However, the results suggest that the supplementation previously mentioned does not affect the shrimp digestive functionality.

### 4.4. Transcriptional Response of Immune-Related Genes

The transcriptional response of immune-related genes is very important to obtain data concerning the shrimp health status [35]. However, it has been reported that low fishmeal diets impair the immune and antioxidant shrimp response due to imbalanced nutrients, high anti-nutritional factors, and fiber content that affect feed intakes, palatability, and digestibility [56,65,66,67]. Previously, we investigated the effects of protein source and level on immune-related genes (Hc, pPO, LGBP), antioxidant capacity (MnSOD), and stress tolerance (HSP60) and suggested that the defense mechanisms were not affected when shrimp were fed diets containing plant-based protein (30–35%) at medium levels [35]. On the other hand, it has been reported that low fishmeal diets have adverse effects on shrimp immune response when the fishmeal was reduced from 250 g/kg to 100 g/kg [68]. Nonetheless, the antioxidant response was modulated without affecting liver and intestine oxidative status when European sea bass (*Dicentrarchus labrax*) was fed low fishmeal supplemented with a DL-Met level 12% below their established requirement [69]. Also, the immune response and antioxidant capacity were improved when white shrimp (*L. vannamei*) [8] and Nile tilapia (*O. niloticus*) [14] were fed a low fishmeal diet with 0.15% MET-MET. Moreover, a study indicated that Nile tilapia (*O. niloticus)* fed low fishmeal diets supplemented with *Spirulina platensis* and *B. amyloliquefaciens* had enhanced immune response and antioxidant capacity, but diminished stress tolerance [21]. The results of this study found the transcriptional responses of genes related to immunity, antioxidant capacity and stress tolerance were improved when shrimp were fed D2, D3, D4, and D5, which is interesting because this indicates that methionine and probiotic supplementation can positively modulate the defense mechanisms, decreasing the affectations caused by low fishmeal feeds. In contrast, the low transcriptional response of immune-related genes observed in shrimp fed D1 could be due to a methionine deficit. However, considering the productive performance, the best dietary treatments could be D3 and D5 without affecting shrimp growth and health.

### 4.5. Digestive Microbiota Analysis and Functional Prediction

The shrimp digestive system hosts microorganism communities dominated by bacteria, but eukaryotic microorganisms may also be present, building a large microbial ecosystem called the microbiota [70]. The shrimp digestive system microbiota influences immunity or resistance, beneficial metabolite production, and nutrient digestion and assimilation [71,72]. Diet is one of the main environmental factors that affects the shrimp digestive system microbiota [73]. This study evaluated the effects of low fishmeal diets on diversity, structure, and relative abundance of prokaryotic and eukaryotic microorganisms from intestine and hepatopancreas of *L. vannamei* using 16S and 18S sequencing. The study results suggested that diversity and structure of the microbiota (prokaryotic and eukaryotic) were not different between dietary treatments. However, beta diversity comparison determined that intestinal prokaryotic microbiota clustered separately from hepatopancreatic prokaryotic microbiota, suggesting a unique ecological niche according to shrimp digestive organ [74]. Therefore, specialized microorganisms in both intestine and hepatopancreas help to improve the energy the host efficiently obtains and the metabolic processes necessary for growth, immune response, nutrients digestion and assimilation [75]. As will be explained below, the study results suggest that the abundance of both prokaryotic and eukaryotic microorganism could increase and help shrimp to efficiently use nutrients from low fishmeal diets for metabolic processes necessary for obtaining energy, growth, immunity, digestion, and nutrition.

According to the abundance of prokaryotes in all intestines and hepatopancreas samples, the most prevalent phylum was Proteobacteria while the Actinobacteria phylum had a higher presence in hepatopancreas samples. These results are in accordance with a previous study [36], and it has also been described that these phyla were dominant in juvenile and adult shrimp (*Penaeus monodon*) [76]. *Pseudoalteromonas* belonging to the Proteobacteria phylum, is a digestive enzyme (proteases, amylases, β-galactosidases, and phospholipases) producing microorganism that contributes to the nutrient digestion of shrimp [77,78,79]; it has also been reported that it contributes to polyunsaturated fatty acid (PUFA) and short-chain fatty acid (SCF) synthesis [80,81]. *Ruegeria*, belonging to the Proteobacteria phylum, produces triesterase activity, contributing to host digestive processes as well as antibacterial activity against *Vibrio anguillarum* [82]. *Demequina* belonging to the Actinobacteria phylum can produce α-amylase, xylanase, and cellulase, which are involved in carbohydrate absorption and utilization [83,84,85,86]. According to chemotaxonomic and genomics profiles, *Lysinimicrobium* can be considered a subjective synonym of *Demequina* [87] and consequently could have the same contributions to digestive enzyme activity and carbohydrate digestion. The role of prokaryotic microorganisms in the shrimp digestive system are closely related to function predictions in the KEGG database, which were amino acid, carbohydrate, cofactors, and vitamins metabolism as well as cell motility. The enzyme digestive production and essential fatty acid synthesis indicate the beneficial role of the microbiota in the health and immune stimulation of the shrimp digestive system. Moreover, cell motility processes such as bacterial chemotaxis and flagellar assembly, could support the prokaryotic microorganisms in adapting to the host digestive environment [88]. Therefore, low fishmeal diets, supplemented with 0.19% MET-MET or 0.06% MET-MET plus probiotic (*B. amyloliquefaciens*) would help the beneficial roles of prokaryotic communities on the shrimp digestive system.

It was difficult to characterize the eukaryotic microorganism abundances with 18S gene sequencing, because a great amount of *L. vannamei* DNA was detected in this study. Furthermore, it is common in fecal DNA metabarcoding that microeukaryotes’ DNA in feces is degraded more rapidly than the host´s DNA and additional steps such as cleavage or blocking host DNA using restriction enzymes or blocking primers is necessary before, or after amplification [89]. Nevertheless, the study results provide insight into the microeukaryotic communities of the shrimp digestive system. The SAR phylum was the most abundant in shrimp intestine fed low fishmeal diets. *Aurantiochytrium* had higher abundance in shrimp intestine fed D3 while *Aplanochytrium* was most abundant in shrimp intestine fed D5. Additionally, *Ebria* was the most prevalent in shrimp intestine fed D1. *Aurantiochytrium* is an eicosapentaenoic acid (EPA) producer [90] whereas *Aplanochytrium* produces docosahexaenoic acid (DHA) [91]. These fatty acids play an essential role in promoting the growth or boosting of aquatic organisms’ immunity; also, EPA and DHA help aquatic organisms reduce inflammatory factors and reduce inflammation [90]. On the other hand, *Ebria* has an internal solid siliceous skeleton, which could result in poor digestion by aquatic organisms [92], which could lower the growth performance of shrimp fed D1. The eukaryotic microorganisms live as commensals or mutualists within the digestive system and tissues of marine invertebrates, but is probable that they originated from coastal water that was used to irrigate the shrimp pond, as previously described [93]. However, the use of diets that are environmentally friendly and supplemented with additives and probiotics could control the harmful microalgae growth in the shrimp digestive system and its environment [94]. This agrees with the study results, which indicate that a low fishmeal diet supplemented with 0.13% DL-MET increased *Ebria* abundance affecting productive performance and health of shrimp, while diets supplemented with 0.19% MET-MET or 0.06% MET-MET plus 0.10% *B. amyloliquefaciens* would help the beneficial roles of eukaryotic communities in the shrimp digestive system.

## 5. Conclusions

Fishmeal could be partially replaced up to 50% by SBM and PBM in shrimp feed supplemented with MET-MET and/or *B. amyloliquefaciens* CECT 5940 without adverse effects on growth performance. In comparison with control diet, shrimp fed reduced fishmeal diets with 0.19% MTE-MET and 0.06% MET-MET plus 0.10% *B. amyloliquefaciens* CECT 5940 had 43% and 40% more final biomass, respectively. Moreover, shrimp fed reduced fishmeal diets supplemented with MET-MET and/or *B. amyloliquefaciens* CECT 5940 had better hepatopancreas epithelial cell height, digestive enzyme activity, transcriptional response of immune-related genes, and beneficial microbiota for the digestive system. Further research is necessary to investigate graduated levels of methionine sources and synergistic effects of amino acids and probiotics in reduced fishmeal diets for farmed shrimp. Also, it would be interesting to evaluate how SBM and PBM influence the fatty acid composition of reduced fishmeal diets and shrimp muscle quality. Nevertheless, this information could be interesting to develop low fishmeal feeds for aquaculture without affecting the growth and welfare of aquatic organisms.

## Figures and Tables

**Figure 1 animals-13-00043-f001:**
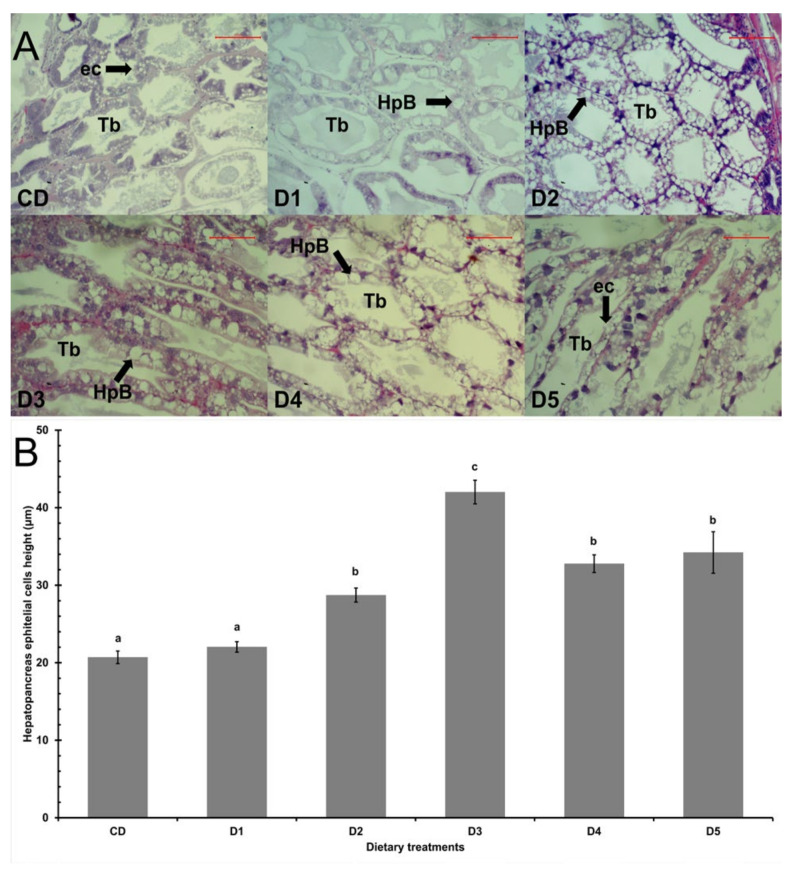
Hepatopancreas histology of shrimp *L. vannamei* fed control diet and 50% fishmeal replacement diets. (**A**) Light micrographs of longitudinal sections (4 μm) of hepatopancreas stained with hematoxylin and eosin, showing a well-organized structure. Arrows indicate normal structures of tubule epithelial cells including secretory (B-cells) cells. Scale bar: 100 μm. (**B**) Hepatopancreas epithelial height cells of shrimp. Data are presented as mean ± SE, values with different letters are significantly different (*p* < 0.05). Abbreviations: Tubules (Tb), epithelial cells (ec), and B-cells (HpB). CD (200 g/Kg FM), D1 (0.13% DL-MET), D2 (0.06% MET-MET), D3 (0.19% MET-MET), D4 (0.13% DL-MET plus 0.10% *B. amyloliquefaciens* CECT 5940) and D5 (0.06% MET-MET plus 0.10% *B. amyloliquefaciens* CECT 5940).

**Figure 2 animals-13-00043-f002:**
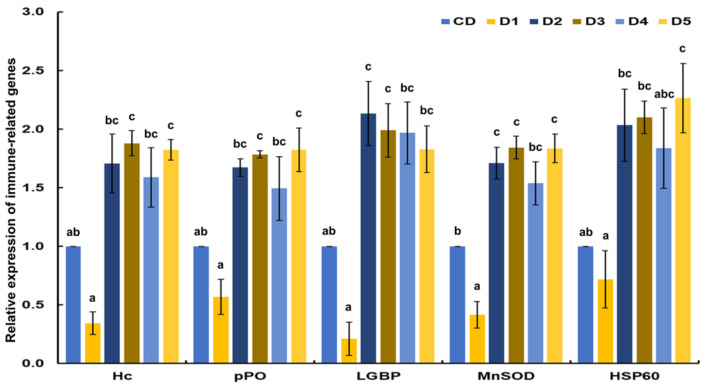
Transcriptional response of immune-related genes of shrimp fed control diet and 50% fishmeal replacement diets. (**A**) *hemocyanin* (Hc), (**B**) *prophenoloxidase* (pPO), (**C**) *lipopolysaccharide-and β-glucan-binding protein* (LGBP), (**D**) *cytosolic manganese superoxide dismutase* (MnSOD), and (**E**) *heat shock protein 60* (HSP60). Data are presented as mean ± SE, values with different letters are significantly different (*p* < 0.05). Abbreviations: CD (200 g/Kg FM), D1 (0.13% DL-MET), D2 (0.06% MET-MET), D3 (0.19% MET-MET), D4 (0.13% DL-MET plus 0.10% *B. amyloliquefaciens* CECT 5940) and D5 (0.06% MET-MET plus 0.10% *B. amyloliquefaciens* CECT 5940).

**Figure 3 animals-13-00043-f003:**
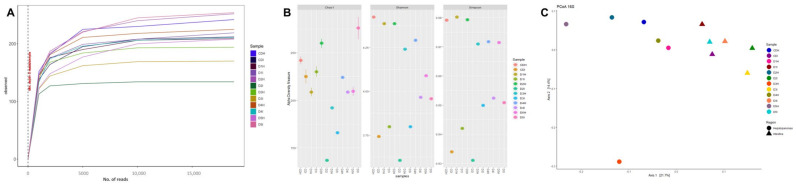
16S rRNA gene sequencing of intestines and hepatopancreas from shrimp fed control diet and 50% fishmeal replacement diets. (**A**) Rarefaction curve, (**B**) Alpha diversity, and (**C**) Microbiota structure visualized using principal coordinate analysis (PCoA) plots. Abbreviations: Intestine (I) and hepatopancreas (H). CD (200 g/Kg FM), D1 (0.13% DL-MET), D2 (0.06% MET-MET), D3 (0.19% MET-MET), D4 (0.13% DL-MET plus 0.10% *B. amyloliquefaciens* CECT 5940) and D5 (0.06% MET-MET plus 0.10% *B. amyloliquefaciens* CECT 5940).

**Figure 4 animals-13-00043-f004:**
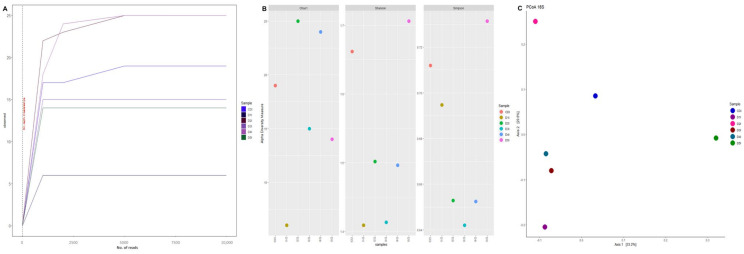
18S rRNA gene sequencing of intestines from shrimp fed control diet and 50% fishmeal replacement diets. (**A**) Rarefaction curve, (**B**) Alpha diversity, and (**C**) Microbiota structure visualized using principal coordinate analysis (PCoA) plots. Abbreviation: Intestine (I). CD (200 g/Kg FM), D1 (0.13% DL-MET), D2 (0.06% MET-MET), D3 (0.19% MET-MET), D4 (0.13% DL-MET plus 0.10% *B. amyloliquefaciens* CECT 5940) and D5 (0.06% MET-MET plus 0.10% *B. amyloliquefaciens* CECT 5940).

**Figure 5 animals-13-00043-f005:**
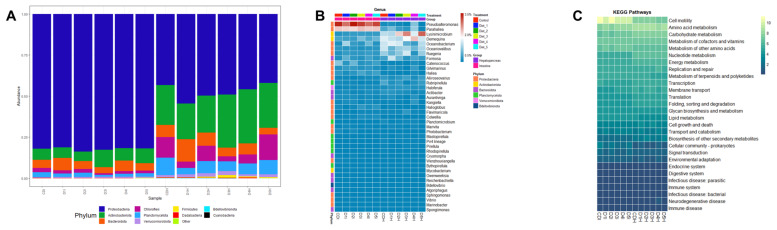
Prokaryotic microbiota of intestines and hepatopancreas from shrimp fed control diet and 50% fishmeal replacement diets. (**A**) Top ten of phyla abundance. (**B**) Heatmap analysis of top 40 genera, and (**C**) Heatmap analysis of function prediction based on KEGG pathways analysis. Abbreviations: intestine (I) and hepatopancreas (H). CD (200 g/Kg FM), D1 (0.13% DL-MET), D2 (0.06% MET-MET), D3 (0.19% MET-MET), D4 (0.13% DL-MET plus 0.10% *B. amyloliquefaciens* CECT 5940) and D5 (0.06% MET-MET plus 0.10% *B. amyloliquefaciens* CECT 5940).

**Figure 6 animals-13-00043-f006:**
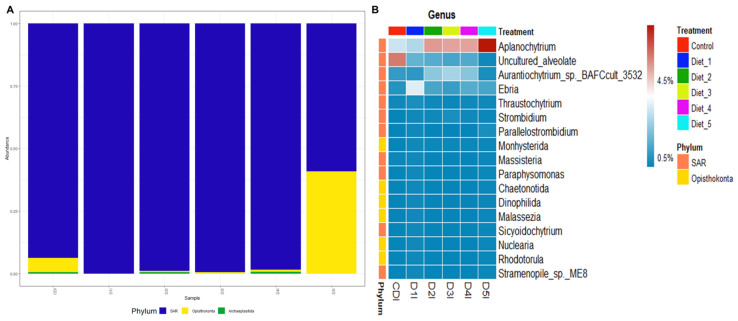
Eukaryotic microbiota of intestines from shrimp fed control diet and 50% fishmeal replacement diets. (**A**) Phyla abundance. (**B**) Heatmap analysis of genera abundance. Abbreviation: intestine (I). SAR (Stramenopiles, Alveolate y Rhizaria). CD (200 g/Kg FM), D1 (0.13% DL-MET), D2 (0.06% MET-MET), D3 (0.19% MET-MET), D4 (0.13% DL-MET plus 0.10% *B. amyloliquefaciens* CECT 5940) and D5 (0.06% MET-MET plus 0.10% *B. amyloliquefaciens* CECT 5940).

**Table 1 animals-13-00043-t001:** Ingredients and proximate composition of the experimental diets (g/kg dry weight).

Ingredients (g/kg Dry Weight)	CD	D1	D2	D3	D4	D5
Fishmeal ^1^	200.0	100.0	100.0	100.0	100.0	100.0
Soybean meal ^2^	277.1	319.3	316.1	316.8	319.7	316.2
Poultry by-product meal ^3^	40.0	80.0	80.0	80.0	80.0	80.0
Wheat flour ^4^	426.9	434.5	438.1	436.2	433.1	437
Soy lecithin ^5^	35.0	35.0	35.0	35.0	35.0	35.0
Fish oil ^6^	15.0	15.0	15.0	15.0	15.0	15.0
Pellet binder ^7^	3.0	3.0	3.0	3.0	3.0	3.0
Antioxidants (Calcium propionate) ^8^	1.0	1.0	1.0	1.0	1.0	1.0
Minerals ^9^	1.0	1.0	1.0	1.0	1.0	1.0
Vitamins ^10^	1.0	1.0	1.0	1.0	1.0	1.0
Mono-dicalcium phosphate ^11^	0.0	2.7	2.7	2.7	2.7	2.7
DL-Methionine ^12^	0.0	1.3	0.0	0.0	1.3	0.0
Met-Met ^13^	0.0	0.0	0.6	1.9	0.0	0.6
Biolys 77 ^14^	0.0	4.9	5.0	5.0	4.9	5.0
L-Threonine ^15^	0.0	1.2	1.3	1.3	1.2	1.3
L-Tryptophan ^16^	0.0	0.1	0.2	0.2	0.1	0.2
*B. amyloliquefaciens* CECT 5940 ^17^	0.0	0.0	0.0	0.0	1.0	1.0
Total (g)	1000.0	1000.0	1000.0	1000.0	1000.0	1000.0
Nutritional composition (%)						
Dry matter	90.4	90.4	90.4	90.4	90.4	90.4
Crude protein	35.0	32.9	32.8	32.9	32.9	32.8
Crude lipid	8.53	8.51	8.51	8.51	8.51	8.51
Ash	5.16	4.86	4.86	4.86	4.86	4.86
Methionine	0.7	0.7	0.7	0.9	0.7	0.7
Lysine	2.1	2.1	2.1	2.1	2.1	2.1
Phosphorus	0.8	0.8	0.8	0.8	0.8	0.8
Gross energy (kcal/kg)	4.54	4.50	4.50	4.50	4.50	4.50
Water stability	0.95 ± 0.002 ^ab^	0.95 ± 0.006 ^ab^	0.94 ± 0.005 ^a^	0.95 ± 0.005 ^ab^	0.96 ± 0.003 ^b^	0.96 ± 0.003 ^b^

^1, 6^ Alimar S.A de C.V. (Cd. Obregón, Sonora, México); ^2^ COLPAC (Navojoa, Sonora, México); ^3, 7–11^ ARY Agroindustrial S.A. de C.V. (Cd. Obregón, Sonora, México); ^4^ MUNSA Molinos S.A de C.V (Cd. Obregón, Sonora, México); ^5^ COLPAC (Navojoa, Sonora, México); ^12–16^ EVONIK México S.A. de C.V. (Tlalpan, CDMX, México). Water stability values are mean ± SEM of three replicates, and values with different letters are significantly different (*p* < 0.05).

**Table 2 animals-13-00043-t002:** Specific primers used for transcriptional response of immune related genes of *L. vannamei*.

Gene	Forward/Reverse Sequence	Amplicon Length (bp)	Efficiency (%)	Correlation Coefficient (R^2^)	GenBank Accession Number
*β-actin*	5′-CCACGAGACCACCTACAAC-3′ 5′-AGCGAGGGCAGTGATTTC-3′	142	91	0.99	AF300705
*hemocyanin* (Hc)	5′-GTCTTAGTGGTTCTTGGGCTTGTC-3′ 5′-GGTCTCCGTCCTGAATGTCTCC-3′	124	98	0.98	X82502
*prophenoloxidase* (proPO)	5′-CGGTGACAAAGTTCCTCTTC-3′ 5′-GCAGGTCGCCGTAGTAAG-3′	122	99	0.99	AY723296
*lipopolysaccharide- and β-glucan-binding protein* (LGBP)	5′-CCATGTCCGGCGGTGGAA-3′ 5′-GTCATCGCCCTTCCAGTTG-3′	122	110	0.99	EU102286
*cytosolic manganese superoxide dismutase* (cytMnSOD)	5′-TGTTGCACAAGCCATTGACGA-3′ 5′-CCAGCCAGAGCCTTTCACTCC-3′	94	90	0.98	DQ005531
*heat shock protein 60* (HSP60)	5′-ATTGTCCGCAAGGCTATC-3′ 5′-ATCTCCAGACGCTTCCAT-3′	102	103	0.99	FJ710169

**Table 3 animals-13-00043-t003:** Effect of experimental diets on growth performance of *Litopenaeus vannamei*.

Diets	Initial Weight (g)	Final Weight (g)	Weekly Weight Gain (g/week)	Specific Growth Rate (%/day)	Survival Rate (%)	Final Biomass (g)	Feed Intake (g)	Feed Conversion Rate
CD	0.28 ± 0.01 ^a^	5.00 ± 0.10 ^ab^	0.59 ± 0.01 ^ab^	5.17 ± 0.08	77.50 ± 5.95	77.64 ± 6.78 ^ab^	145.19 ± 11.06 ^ab^	1.87 ± 0.02
D1	0.31 ± 0.02 ^a^	4.60 ± 0.38 ^a^	0.54 ± 0.05 ^a^	4.81 ± 0.28	73.75 ± 6.57	67.95 ± 8.42 ^a^	131.08 ± 17.28 ^a^	1.94 ± 0.16
D2	0.29 ± 0.02 ^a^	5.34 ± 0.26 ^ab^	0.63 ± 0.03 ^ab^	5.21 ± 0.11	86.25 ± 7.74	93.21 ± 12.10 ^ab^	160.06 ± 14.88 ^ab^	1.75 ± 0.09
D3	0.32 ± 0.02 ^a^	5.84 ± 0.25 ^b^	0.69 ± 0.03 ^b^	5.20 ± 0.18	95.00 ± 3.54	111.11 ± 7.02 ^b^	185.63 ± 7.15 ^ab^	1.68 ± 0.06
D4	0.28 ± 0.02 ^a^	5.28 ± 0.10 ^ab^	0.63 ± 0.01 ^ab^	5.27 ± 0.12	83.75 ± 6.57	88.26 ± 6.64 ^ab^	163.87 ± 12.81 ^ab^	1.86 ± 0.06
D5	0.32 ± 0.01 ^a^	5.98 ± 0.21 ^b^	0.71 ± 0.03 ^b^	5.22 ± 0.13	91.25 ± 3.75	108.94 ± 4.39 ^b^	190.83 ± 10.46 ^b^	1.75 ± 0.04
** *p-* ** **value**	0.3196	0.0063	0.0109	0.4152	0.1507	0.0069	0.0285	0.3419

Values are mean ± SEM of three replicates, and values in the same row with different letters are significantly different (*p <* 0.05).

**Table 4 animals-13-00043-t004:** Effect of experimental diets on digestive enzyme activity of *Litopenaeus vannamei*.

Digestive Enzyme Activity [U/mg]	Dietary Treatment						
	CD	D1	D2	D3	D4	D5	*p* Value
Hepatopancreas							
Amylase	75.09 ± 3.16 ^a^	61.78 ± 1.87 ^a^	94.64 ± 9.12 ^a^	369.48 ± 4.73 ^b^	165.30 ± 70.30 ^a^	154.90 ± 28.38 ^a^	0.0010
Protease	0.14 ± 0.01 ^a^	0.16 ± 0.01 ^a^	0.42 ± 0.10 ^ab^	1.05 ± 0.21 ^b^	0.41 ± 0.22 ^ab^	0.33 ± 0.13 ^a^	0.0073
Lipase	1.32 ± 0.12 ^a^	0.80 ± 0.10 ^a^	1.05 ± 0.07 ^a^	5.53 ± 0.38 ^b^	1.86 ± 0.67 ^a^	2.70 ± 1.00 ^a^	0.0003
Intestine							
Amylase	249.49 ± 8.91	202.36 ± 36.41	279.44 ± 48.10	291.38 ± 83.74	385.98 ± 104.15	405.49 ± 9.7	0.2044
Protease	16.80 ± 0.26	12.64 ± 3.14	18.18 ± 5.09	18.84 ± 2.06	22.62 ± 6.49	30.59 ± 2.1	0.0815
Lipase	1.48 ± 0.11	1.11 ± 0.25	1.61 ± 0.28	1.86 ± 0.40	2.39 ± 0.9	2.28 ± 0.08	0.3943

Values are mean ± SEM of three replicates, and values in the same row with different letters are significantly different (*p <* 0.05).

## Data Availability

The datasets presented in this study can be found in online repositories. The names of the repository/repositories and accession number(s) can be found below: NCBI Sequence Reads (SRA) under BioProject: PRJNA872278.

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
