# Peer review of "Dietary Use of Methionine Sources and Bacillus amyloliquefaciens CECT 5940 Influences Growth Performance, Hepatopancreatic Histology, Digestion, Immunity, and Digestive Microbiota of Litopenaeus vannamei Fed Reduced Fishmeal Diets"

_animals, 2022, doi:10.3390/ani13010043_

Round 1
Author Response
First, thanks for your observations, suggestions, and question. Everything was beneficial for manuscript improvement. All changes are highlighted in yellow color.
Reply to your specific comments:
- In Table 1, it has flaws in the experimental design.
--The crude protein content was 35% in the control group and about 32.9% in other five groups, which was not the same as the author described in abstract and in Line 114: “Six isonitrogenous (about 33% crude protein)”.
R: Thanks very much for the observation. You are right, the description as “six isonitrogenous” diets can cause confusion to readers. Therefore, we changed the description only as six experimental diet (Lines 33 and 122).
--In the diet formula, the fishmeal was replaced not only by plant ingredients including soy flour and wheat flour but also by poultry meal. Meanwhile, the fish oil and essential fatty acids could be changed in the control and other five groups, which might also have influences on the experimental results. In this case, it is not suitable for only mentioned the methionine balance and microbial composition. The effects of these different diet design should be added and detailed described in the material and methods and discussion.
R: Totally agree with your suggestion. It isn’t suitable to mention that it could be replaced 50% fishmeal by additive supplementation (methionine source and probiotic) without considering the nutritional value of alternative protein source. Therefore, we added information regarding soybean meal and poultry by-product meal in simple summary (Lines 20 to 22), abstract (Lines 30 to 34, 49 to 56), introduction (Lines 79 to 79, 113 to 114), materials and methods (Lines 123 to 133), discussion (Line 444 to 451), and conclusion (Lines 601 to 603, 611 to 612).
--Considering the above reasons, it is not suitable for mentioning in the abstract, results and conclusions that it could be replaced 50% fishmeal by supplementation of the methionine and microbial composition.
R: The same answer as in the previous question.
--In the nutritional composition, the crude lipid of the six experimental diets should be added in the Table 1.
R: Crude lipid was added in the nutritional composition of Table 1.
- Line 117-120: As the authors mentioned, there are different water solubility ratio between DL-Met and AQUAVI®. Considering the slowly feeding habits of the shrimp, it is necessary to measure and compare the water solubility ratio of Met (AQUAVI®) in feed pallets and add it in the results so as to comprehensive evaluation of the Met and/or microbial composition addition effects.
R: Water solubility was not determined in this study. We believe that is a reason due to previous reports as we mentioned in introduction (Lines 88 to 90) and discussion (Lines 432 to 434). Therefore, we added a research perspective in discussion (Lines 434 to 436).
- Formatting correction
--Line 137: m3 should be changed to m3.
R: The change was made.(Line 155)
--Line 153: there should be a space in “studies[24, 25, 26]”.
R: A space was added (Line 171).
--Line 290: D2, D3 y D5, “y”?
R: The mistake was corrected (Line 325).
Reviewer 2 Report
The paper ¨Dietary use of methionine sources and Bacillus amyloliquefaciens CECT 5940 influences growth performance, hepatopancreatic histology, digestion, immunity, and digestive microbiota of Litopenaeus vannamei fed reduced fishmeal diets¨ presents novel information on reducing the use of FM in shrimp aquafeeds. Nevertheless, some observations must be addressed.
Major remarks:
Why if throughout the text it is mentioned that two factors are being tested, a two-way ANOVA has not been performed. A two-way ANOVA should be performed on results such as organism performance (at least), in order to make a coherent decision regarding what has been tested.
Regarding gene expression:
- What method was used to determine the expression of the genes (absolute, relative, etc.)
-what conditions were used to synthesize the cDNA and in the expression of the genes
-RT is the machine, what is determined is a quantitative PCR (qPCR), modify it in the text
-In the name of the primers they are not determinative nor do they provide necessary information (each author names them as he wants), remove it from table 2
-Add in table 2, amplicon sizes in base pairs (bp), reaction efficiencies (E), and Pearsons coefficients of determination (R2)
-Use the proper nomenclature to differentiate between proteins and genes, by convention proteins are in upper case while genes are in lower case and italics (make changes throughout the text)
What is the difference between alpha and beta diversity? if you want to compare different sites (in this case organs), it is recommended to use beta diversity
In relation to the histology images, make an in-depth description of what is being represented (clearly explain what the arrows are pointing to)
A thorough revision of the English is recommended, there are unfinished ideas and the exaggerated use of connectors that are repeated throughout the text such as ¨and¨
Minor remarks:
· The term aquafeeds is used excessively at the same time that it is used in a general way, from the way it is used it seems that it is for all species and the authors have only worked with shrimp (clarify for which species it is mentioned)
· Throughout the text common names of the species are mentioned without mentioning the scientific name, add all the scientific names.
L21-22: improve editing, the idea is not clear
L67-71: go deeper with the examples, the paragraph is very general
L89-90: unfinished ideas
L94- Edition
L129- cite the source
L143-148 - improve editing, the idea is not clear
L304-317 - the paragraph is methodology, move to the corresponding section
· The important thing is the components of the diet, not the brand, investor and put the commercial within the parentheses
· Table 1. (a) edit the header as the treatments are confused. (b) Besides, why is starch in nutritional content?? (IT IS AN INGREDIENT). (c) Include the companies from which the different ingredients have been obtained (it is necessary in nutritional experiments). (d) What is the purpose of including sums of amino acids in the table, what new information do they provide? what biological utility can it have? throughout the text they do not mention this purpose (remove all the extra information that does not contribute anything to the work).
· Why is it mentioned that the shrimp were collected (as they would have been caught) if they were bought from a farm (clarify ideas)
· 0.3 gr shrimp are already juveniles?
· Review and correct final weight formula
· In relation to histology, how many cuts have been made per sample? How many readings have been made of each sample? were done in triplicate? before making a decision of what has been found
· Add the p value in the tables
· Standardize the use of units, in some cases the % is together and in others it is separated from the unit
Author Response
Author's Reply to the Review Report (Reviewer 2)
First, thanks for your observations, suggestions, and question. Everything was beneficial for manuscript improvement. All changes are highlighted in yellow color.
Regarding major remarks.
Why if throughout the text it is mentioned that two factors are being tested, a two-way ANOVA has not been performed. A two-way ANOVA should be performed on results such as organism performance (at least), in order to make a coherent decision regarding what has been tested.
R: You're absolutely right, and we apologize for creating confusion as a result of incorrect writing. Our experimental design was one-factor due to the study purpose was to evaluate the effect of FM replacement by SBM and PBM in diets supplemented with DL-Met, MET-MET (AQUAVI®), Bacillus amyloliquefaciens CECT 5940 (ECOBI-OL®) and their combinations on growth performance and health of juvenile Litopenaeus vannamei. The redaction was corrected throughout the text to avoid confusion.
Regarding gene expression:
- What method was used to determine the expression of the genes (absolute, relative, etc.)
R: Method to determine gene expression was added in Line 213 and 214.
-what conditions were used to synthesize the cDNA and in the expression of the genes
R: Condition to cDNA synthesis were added in Lines 204 – 206.
-RT is the machine, what is determined is a quantitative PCR (qPCR), modify it in the text
R: The text was modified in Lines 207, 209 and 210.
-In the name of the primers they are not determinative nor do they provide necessary information (each author names them as he wants), remove it from table 2
R: Primer names were removed from Table 2.
-Add in table 2, amplicon sizes in base pairs (bp), reaction efficiencies (E), and Pearsons coefficients of determination (R2)
R: The requested information was added in Table 2. Also, some modifications were included in amplicon length.
-Use the proper nomenclature to differentiate between proteins and genes, by convention proteins are in upper case while genes are in lower case and italics (make changes throughout the text)
R: Gene names were changed throughout the text according to conventional nomenclature.
What is the difference between alpha and beta diversity? If you want to compare different sites (in this case organs), it is recommended to use beta diversity
R: Alpha diversity refers to diversity on a single sample, describing the species diversity (richness) within a functional community. Beta diversity, on the other hand, describes the amount of differentiation between different sites. Therefore, results description regarding microbiota was modified from Lines 345 to 350. Also, in discussion section we added information in Lines 535 to 543.
In relation to the histology images, make an in-depth description of what is being represented (clearly explain what the arrows are pointing to)
R: Figure legend was rewritten for better comprehension.
A thorough revision of the English is recommended, there are unfinished ideas and the exaggerated use of connectors that are repeated throughout the text such as ¨and¨
R: English language was revised to avoid exaggerated use of connector as well as to clarify ideas throughout the text.
Minor remarks:
- The term aquafeeds is used excessively at the same time that it is used in a general way, from the way it is used it seems that it is for all species and the authors have only worked with shrimp (clarify for which species it is mentioned)
R: In the revised version of manuscript, the term aquafeed was only used in a general way throughout the text.
- Throughout the text common names of the species are mentioned without mentioning the scientific name, add all the scientific names.
R: All scientific names were added to manuscript.
L21-22: improve editing, the idea is not clear
R: The idea was clarified, and edition was improved (Lines 22 to 25).
L67-71: go deeper with the examples, the paragraph is very general
R: Examples were added (Lines 71 to 79)
L89-90: unfinished ideas
R: Ideas were finished, and edition was improved (Lines 96 to 98)
L94- Edition
R: The edition was realized (Lines 101 to 102).
L129- cite the source
R: Source was cited (Line 147).
L143-148 - improve editing, the idea is not clear
R: Edition was improved, and idea was clarified (Lines 161 to 167).
L304-317 - the paragraph is methodology, move to the corresponding section
R: The paragraph was moved to corresponding section (256 to 269).
- The important thing is the components of the diet, not the brand, investor and put the commercial within the parentheses
R: All changes were made throughout the text.
- Table 1. (a) edit the header as the treatments are confused. (b) Besides, why is starch in nutritional content?? (IT IS AN INGREDIENT). (c) Include the companies from which the different ingredients have been obtained (it is necessary in nutritional experiments). (d) What is the purpose of including sums of amino acids in the table, what new information do they provide? what biological utility can it have? throughout the text they do not mention this purpose (remove all the extra information that does not contribute anything to the work).
R: Table 1 was modified according to your suggestion for better comprehension. All extra information that does not contribute anything to the work was removed. Thanks again for your observations.
- Why is it mentioned that the shrimp were collected (as they would have been caught) if they were bought from a farm (clarify ideas)
R: Collected word was changed for obtained word (Line 145).
- 0.3 gr shrimp are already juveniles?
R: In shrimp farming industry there are two main systems for the management, maintenance, and growing of organisms. When the shrimp are in the nursery system, they are considered postlarvae. When the shrimp are in the grow-out system, they are considered juveniles. For example, in the central USA, water temperatures in production ponds are suitable for approximately 120 to 140 days per year. To achieve marketable sizes of 20 to 30 g within that limited growing period requires a sufficient supply of juveniles of approximately 0.3 to 0.5 g mean individual weight to stock production ponds at 60 000 to 100 000 juveniles/ha.
Rodríguez-Olague, D., Ponce-Palafox, J. T., Castillo-Vargasmachuca, S. G., Arámbul-Muñoz, E., Raúl, C., & Esparza-Leal, H. M. (2021). Effect of nursery system and stocking density to produce juveniles of whiteleg shrimp Litopenaeus vannamei. Aquaculture Reports, 20, 100709.
Tidwell, J. H., & D'Abramo, L. R. (2000). Grow‐out Systems‐Culture in Temperate Zones. Freshwater prawn culture: the farming of Macrobrachium rosenbergii, 177-186.
- Review and correct final weight formula
R: Final weight formula was reviewed and corrected (Line 172).
- In relation to histology, how many cuts have been made per sample? How many readings have been made of each sample? were done in triplicate? before making a decision of what has been found
R: An average of 50 cuts by sample were made from stomach, hepatopancreas and intestine. However, due to staining alteration in stomach and intestine sample we decided to analyze the hepatopancreas histology. An average of 15 hepatopancreas cuts by samples were visualized, and an average of 317 readings were realized to measure hepatopancreas cell height. (Here share a link).
- Add the p value in the tables
p values were added in the tables.
- Standardize the use of units, in some cases the % is together and in others it is separated from the unit
R: The use of units was standardized throughout the text.
Round 2
Reviewer 1 Report
As the authors mentioned, there are different water solubility ratio between DL-Met and AQUAVI®. Considering the slowly feeding habits of the shrimp, it is necessary to measure and compare the water solubility ratio of Met (AQUAVI®) in feed pallets and add it in the results so as to comprehensive evaluation of the Met and/or microbial composition addition effects.
Author Response
First, thank you very much for your suggestion. Your review and contribution were very valued for manuscript improvement.
The water stability calculation in feed pellet was added in materials and methods section (lines 135 to 137). The water stability values were added at the bottom of Table 1. Also, a small note was added in discussion section (lines 422 to 424).
Reviewer 2 Report
The authors have completed all the suggested comments
Author Response
Thank you very much for your suggestion and observations. Your review and contribution were very valued for manuscript improvement.